# The source of discrepancies in aerosol–cloud–precipitation interactions between GCM and A-Train retrievals

Takuro Michibata[1,2], Kentaroh Suzuki[3], Yousuke Sato[4], and Toshihiko Takemura[2]

[1]Department of Earth System Science and Technology, Kyushu University, Fukuoka, Japan
[2]Research Institute for Applied Mechanics, Kyushu University, Fukuoka, Japan
[3]Atmosphere and Ocean Research Institute, The University of Tokyo, Chiba, Japan
[4]RIKEN Advanced Institute for Computational Science, Hyogo, Japan

*Correspondence to:* T. Michibata (michibata@riam.kyushu-u.ac.jp)

**Abstract.** Aerosol–cloud interactions are one of the most uncertain processes in climate models due to their nonlinear complexity. A key complexity arises from the possibility that clouds can respond to perturbed aerosols in two opposite ways, as characterized by the traditional "cloud lifetime" hypothesis and more recent "buffered system" hypothesis. Their importance in climate simulations remains poorly understood. Here we investigate the response of the liquid water path (LWP) to aerosol
perturbations for warm clouds from the perspective of general circulation model (GCM) and A-Train remote sensing, through process-oriented model evaluations. A systematic difference is found in the LWP response between the model results and observations. The model results indicate a near-global uniform increase of LWP with increasing aerosol loading, while the sign of the response of the LWP from the A-Train varies from region to region. The satellite-observed response of the LWP is closely related to meteorological/macrophysical factors, in addition to the microphysics. The model does not reproduce this variabil-
ity of cloud susceptibility (i.e., sensitivity of LWP to perturbed aerosols) because the parameterization of the autoconversion process assumes only suppression of rain formation in response to increased cloud droplet number, and does not consider macrophysical aspects that serve as a mechanism for the negative responses of the LWP via enhancements of evaporation and precipitation. Model biases are also found in the precipitation microphysics, which suggests that the model generates rainwater readily even when little cloud water is present. This essentially causes projections of unrealistically frequent and light rain,
with high cloud susceptibilities to aerosol perturbations.

## 1 Introduction

Aerosol particles play an important indirect role in the climate system by modifying cloud micro- and macrophysical properties, which is referred to as aerosol–cloud interactions (Twomey, 1977; Albrecht, 1989). An increase in aerosols supplies more numerous cloud condensation nuclei, resulting in numerous and smaller cloud droplets leading to brighter clouds, which is
known as the "albedo effect" (Twomey, 1977). Smaller cloud droplets suppress the onset of precipitation in warm clouds due to the less efficient collision–coalescence process, resulting in a longer cloud lifetime, which is known as the "lifetime effect" (Albrecht, 1989). There has been much discussion about the climatic impacts of aerosol-induced modulation of water clouds, which are particularly sensitive to aerosol perturbations (e.g., Pincus and Baker, 1994; Bréon et al., 2002; Penner et al., 2006;

Lebsock et al., 2008; Quaas et al., 2009; Terai et al., 2012, 2015). However, quantitative estimates of radiative forcing with regard to aerosol–cloud–precipitation–climate interactions remain uncertain as reported in the Fifth Assessment Report of the Intergovernmental Panel on Climate Change (2013).

One of the most important factors that quantify the magnitude of aerosol–cloud interactions is the response of the cloud liquid water path (LWP) to aerosol perturbations. This factor also characterizes aerosol impacts on the global hydrological cycle through its representation of the aerosol effect on precipitation efficiency. This effect is represented in general circulation models (GCMs) as aerosol-induced changes in rainwater production from cloud water, which are parameterized with a bulk microphysics as the so-called autoconversion process. The water conversion rate by this process ($P_{aut}$) is generally given as a function of the liquid water content ($L_c$) and cloud droplet number concentration ($N_c$) as

$$P_{aut} \sim L_c^{\alpha} \times N_c^{-\beta}, \tag{1}$$

where $\alpha$ and $\beta$ are prescribed constants (e.g., Berry, 1968; Beheng, 1994; Khairoutdinov and Kogan, 2000). The $N_c$ is then somehow related to the aerosol number concentration ($N_a$). In GCMs, Eq. (1) provides the only pathway through which aerosols modulate precipitation formation and, thus, the cloud lifetime. Note that GCMs also partly include the opposing processes (decreasing LWP due to enhancement in evaporation) via the so-called direct and semi-direct effect (e.g., Ackerman et al., 2000; Hansen et al., 1997). Given that rainwater production is always suppressed with increasing $N_c$ according to Eq. (1), GCMs tend to increase the LWP uniformly with increasing $N_c$ for stratiform clouds.

On the other hand, some observational studies have shown two pathways of LWP responses to perturbed aerosols, i.e., both increasing and decreasing tendencies of LWP with increasing aerosols (Matsui et al., 2006; Lebsock et al., 2008; Chen et al., 2014); the mechanisms for these opposing responses cannot be understood by a simple microphysical argument alone, but are likely to relate to macrophysical and meteorological factors as well (e.g., Ackerman et al., 2004; Matsui et al., 2006; Suzuki et al., 2008; Lebsock et al., 2008; Small et al., 2009; Gryspeerdt and Stier, 2012; Chen et al., 2014). Wood (2007) found that there are processes that modify the cloud geometric thickness to aerosol perturbations in such a way that cancels the aerosol indirect effect at sufficiently long timescales. Such a compensation mechanism is currently considered one of the "buffering effects" (Stevens and Feingold, 2009), which generate the opposite result to the original hypotheses of cloud albedo and lifetime effects, for the cloud system as a whole (Rosenfeld et al., 2014; Lebo and Feingold, 2014; Seifert et al., 2015). Despite its critical importance to accurate climate simulations, the operation of this mechanism at the global scale remains poorly understood.

To determine the mechanisms involved in the competition between the "lifetime effect" and "buffering effect", the complexity in aerosol effects on clouds needs to be untangled at a fundamental process-level. For this purpose, GCMs should be evaluated extensively against observations in the context of their process representations, which are key to the aerosol–cloud–precipitation interaction.

In this study, we analyze results from both GCM and A-Train data, with a particular focus on their discrepancies in the key indices of aerosol–cloud interactions relating to fundamental processes. The factors examined are the susceptibilities of cloud optical thickness ($\tau_c$), droplet effective radius ($r_e$), and LWP to $N_c$. To focus directly on the cloud physical parameters, we use

$N_c$ as an aerosol proxy rather than $N_a$ (e.g. Koren and Feingold, 2011). Satellite-based study by Chen et al. (2014) reported that cloud susceptibilities show similar results whether aerosol index, aerosol optical depth (AOD), or $N_c$ are applied as an aerosol proxy (see their supplementary information). Given the fundamental relationship of $\tau_c \propto \mathrm{LWP}/r_e$, the susceptibilities are related as follows (Ghan et al., 2016):

$$\frac{d\ln\tau_c}{d\ln N_c} = -\frac{d\ln r_e}{d\ln N_c} + \frac{d\ln\mathrm{LWP}}{d\ln N_c}, \tag{2}$$

where the first and second terms on the right side of Eq. (2) represent the "albedo effect" and the "lifetime effect", respectively. Equation (2) has the advantage that it can quantify the contributions from the two effects that determine the aerosol impact on cloud radiative properties. As discussed here and also in recent studies (Ghan et al., 2016; Feingold et al., 2016), the two terms in Eq. (2) are related to representations of different processes. This approach makes it easier to understand the mechanisms that determine the resultant magnitude of aerosol indirect forcing in the context of relevant processes (Seinfeld et al., 2016).

The aim of the study is to clarify the fundamental source of uncertainty in process representations of aerosol–cloud–precipitation interactions in GCMs for stratiform and shallow cumulative warm clouds (excluding deep convective thick clouds or ice clouds; see Sect. 2). Given that the aerosol–cloud interaction processes are also influenced by both macrophysics (e.g., environmental conditions, dynamical regime, cloud type) and microphysics, we also place an emphasis on the importance of macrophysics (e.g., Sorooshian et al., 2013; Gryspeerdt et al., 2014; Zhang et al., 2016).

## 2  Data

### 2.1  MIROC-SPRINTARS

A global climate model, "Model for Interdisciplinary Research On Climate (MIROC)" version 5.2 (Watanabe et al., 2010) was used in this study. The interactions of the main tropospheric aerosols (i.e., black carbon, organic matter, soil dust, sea salt, sulfate, and the precursor gases of sulfate) with cloud–precipitation microphysics and radiation–climate effects are incorporated in the aerosol module, "Spectral Radiation-Transport Model for Aerosol Species (SPRINTARS)" (Takemura et al., 2000, 2002, 2005), which is coupled with MIROC (MIROC-SPRINTARS).

The cloud macro- and microphysics framework in MIROC-SPRINTARS is based on a prognostic large-scale condensation scheme, which explicitly considers subgrid-scale variability of clouds (Watanabe et al., 2009). This PDF-based prognostic cloud scheme couples with the ice microphysics scheme proposed by Wilson and Ballard (1999). MIROC-SPRINTARS treats both cloud droplets and ice crystals as a two-moment bulk microphysics scheme (Takemura et al., 2009). The nucleation of cloud droplets is parameterized by the scheme of Abdul-Razzak and Ghan (2000), and the process of cloud-to-rain water conversion is diagnosed based on the Berry (1968) autoconversion scheme. Rainwater is not a prognostic variable in the current version of MIROC-SPRINTARS.

We extracted warm-phase low clouds ($> 273.15$ K in whole cloud layers) from every six hours instantaneous output for five full years; as a result, 1,595,753 warm cloud samples were obtained. The horizontal and vertical resolutions were T42

(approximately 2.8° × 2.8° in latitude and longitude) and 20 layers, respectively. A more detailed description of the model and its settings are documented in Michibata and Takemura (2015).

## 2.2 CloudSat and MODIS

We used the synergistic satellite data sets of the CloudSat and MODIS, which are both part of the A-Train constellation (Stephens et al., 2002, 2008). The data products, 2B-TAU (Polonsky, 2008), 2B-GEOPROF (Marchand et al., 2008), and ECMWF-AUX (Partain, 2007) were used for the period June 2006 to April 2011, i.e., a total of five full years. This facilitated the construction of stable statistics with a horizontal resolution (2.5° grid-boxes in this study) close to the GCM output. We defined cloud layer where the cloud mask value greater than 30 from the 2B-GEOPROF product, which means good/strong echo with high-confidence detection (Marchand et al., 2008). The analysis was restricted to single-layer water clouds; in total, 7,872,426 cloud samples were obtained.

The LWP was derived from the MODIS-retrieved optical thickness and effective radius using the following equation for an adiabatically stratified cloud (Szczodrak et al., 2001):

$$\text{LWP} = \frac{5}{9}\tau_\text{c}\text{r}_\text{e}, \tag{3}$$

$N_c$ was also calculated based on an adiabatic assumption (Wood, 2006) as

$$N_c = \sqrt{2}B^3\Gamma_\text{eff}^{1/2}\frac{\text{LWP}^{1/2}}{r_e^3}, \tag{4}$$

where $B = (3/4\pi\rho_w)^{1/3} = 0.0620 \text{ kg}^{-1/3} \text{ m}$, $\rho_w$ is the density of liquid water, and $\Gamma_\text{eff}$ is the adiabatic rate ($\text{g m}^{-3}\text{ km}^{-1}$) of increase in the liquid water content with height (see also Kubar et al. (2009) for more details of derivation). We note that satellite data inherently include uncertainties stemming from retrieval assumptions, which are not replicated in the model output. Although it could be a part of reason for discrepancies between the model and observations, this would mostly be canceled when susceptibilities of cloud and precipitation to aerosol loading are evaluated by a logarithmic form. This study applies the uncertainty thresholds of $< 5$ and $< 1$ µm for $\tau_c$ and $r_e$ from the 2B-TAU product, respectively, which contributes to reduce the retrieval uncertainty of $N_c$ described above as much as possible (Michibata et al., 2014).

To examine the cloud-to-rain conversion process, the conversion rate ($P_{conv}$) contributed from both autoconversion (collision–coalescence of cloud droplets) and accretion (collision of cloud droplets by raindrops) was derived from the approximation suggested by Stephens and Haynes (2007). This method is established by the continuous collection equation (Pruppacher and Klett, 1997) using observed drop size distributions. $P_{conv}$ was estimated from MODIS LWP and CloudSat mean cloud-layer radar reflectivity $\overline{Z}$ as

$$P_{conv} = c_1 \text{LWP}\,\overline{Z}\,H[Z - Z_c], \tag{5}$$

where $c_1 = \kappa_2/2^6$ is a coefficient from collection kernel (Long, 1974) with $\kappa_2 = 1.9 \times 10^{11} \text{ cm}^{-3}\text{ s}^{-1}$ and sixth moment factor with radar reflectivity. $H[Z - Z_c]$ is the Heaviside step function to exclude the cases that is less than critical radar threshold $Z_c$

of $-15\,\mathrm{dBZ_e}$ for which conversion process is negligible (Matrosov et al., 2004). Although this formulation is based on marine stratocumulus cases from DYCOMS-II measurements (vanZanten et al., 2005), it is applicable for global analysis to study aerosol–cloud interactions (Stephens and Haynes, 2007; Sorooshian et al., 2013) in drizzling light rain cases ($\overline{Z} < 0\,\mathrm{dBZ_e}$). The parameterization and assumptions used in this method (Eq. 5) are also valid for comparison between observations and model simulation (Suzuki and Stephens, 2009). This brings valuable understanding for microphysical conversion processes and its timescales, which matches the scope of our study.

Lower-tropospheric stability (LTS) was derived from ECMWF-AUX product as the difference in potential temperatures between 700 hPa and the surface (Klein and Hartmann, 1993), and is used for a metric of macroscopic thermodynamic conditions.

## 3 Results

### 3.1 Precipitation microphysics

A commonly known problem in GCMs associated with low cloud precipitation microphysics is the timing of precipitation and its frequency/intensity (Suzuki et al., 2013b, 2015). This issue is also related to the magnitudes of the aerosol indirect effect, i.e., dependency of precipitation on $N_c$. As a proxy for this, we use the precipitation susceptibility ($S_p$) metric, defined as

$$S_p = -\frac{d\ln R}{d\ln N_c}, \tag{6}$$

where $R$ is the rain rate. Observed values of $R$ are derived from CloudSat radar reflectivity through the $Z$–$R$ relationship (Comstock et al., 2004), while model values of $R$ are obtained as the large-scale precipitation rate. We apply a threshold of $R > 0.14\,\mathrm{mm\,day^{-1}}$, which is equivalent to a radar reflectivity of $-15\,\mathrm{dBZ_e}$ (Terai et al., 2015), for precipitation in the model to enable a fair comparison with satellite observations. The $S_p$ metric is useful for examining the aerosol impact on precipitation (Sorooshian et al., 2009; Feingold and Siebert, 2009).

Figure 1a shows the behavior of $S_p$ as a function of the LWP obtained from MIROC and A-Train satellite observations. The satellite $S_p$ increases with increasing LWP up to around LWP $\sim 450\,\mathrm{g\,m^{-2}}$; further increases in LWP result in a decrease in $S_p$. This behavior can be interpreted as follows (Sorooshian et al., 2009). $S_p$ is low for a low LWP because clouds cannot generate much rainwater, regardless of the aerosol loading. At a high LWP, $S_p$ is also low because precipitation is dominant, regardless of the aerosol loading, due to the abundant LWP. In other words, the LWP value at which the $S_p$ peaks corresponds to the turning point where the water conversion process shifts from the autoconversion regime to the accretion regime, as suggested by previous studies (Wood et al., 2009; Sorooshian et al., 2009).

On the other hand, the $S_p$ amplitude predicted by MIROC is smaller than the satellite results for a wide range of LWPs, and $S_p$ remains high even after its values peak near LWP $\sim 450\,\mathrm{g\,m^{-2}}$. This is mainly because the autoconversion parameterization assumes a constant dependency on $N_c$ (i.e., $\beta$) regardless of the LWP, as is clear from Eq. (1). This also leads to a significant overestimation of $S_p$ for LWP $< 100\,\mathrm{g\,m^{-2}}$, which means that the model readily generates rainwater even when only a small amount of cloud water is present.

To understand the uncertainty in the conversion process from cloud water to rainwater in more detail, we define a new metric, the "susceptibility of microphysical conversion ($S_{conv}$)" as

$$S_{conv} = -\frac{d \ln P_{conv}}{d \ln N_c}. \tag{7}$$

This metric represents how aerosol burdens suppress rainwater production. In satellite analysis, $P_{conv}$ can be estimated from the method proposed by Stephens and Haynes (2007) as shown in Eq. (5). In the model, $P_{conv}$ is obtained as a native output of the process rate. The method of Stephens and Haynes (2007) was compared with a native model output of the process rate in a global cloud-resolving model (CRM) by Suzuki and Stephens (2009). The study showed that the radar reflectivity is a gross measure of the water conversion time scale, supporting the underlying assumption of Stephens and Haynes (2007). This implies that the $S_{conv}$, which represents the time scale dependency on $N_c$, can be compared between satellite observations and model simulations although absolute values of $P_{conv}$ can be different between them. Although the use of satellite simulators would be helpful for more direct comparison between model and observations, it is left as the subject of future work.

As shown in Fig. 1b, MIROC overestimates $S_{conv}$, particularly in the lower LWP range. This means that the model generates precipitation at a higher frequency, even at low LWPs, compared to observations, which is mainly because the autoconversion in the model is too rapid (Michibata and Takemura, 2015; Suzuki et al., 2015), as described above. Consequently, the probability distribution function (PDF) of the LWP is biased toward lower values because cloud water is depleted quickly by the rapid surface precipitation. Alternatively, it is also possible that the model has biases in the condensation processes, which lead to lower LWP and, thus, result in lower autoconversion rate. These tendencies in the model are strongly related to unrealistically light rain that is too frequent, which is a common problem in GCMs (Stephens et al., 2010), including MIROC.

Besides this, $S_{conv}$ can also be biased from the error of cloud geometric thickness due to insufficient vertical resolution in GCMs. In addition to the microphysical aspects mentioned above, biases in macrophysical structure are also related to model performances, which will be discussed later (cf. Sect. 3.3).

## 3.2 Cloud susceptibilities

The response of cloud liquid water to aerosol perturbations determines the cloud lifetime via the modification of cloud fraction (Albrecht, 1989), and is thus related to global hydrological cycles as well as radiation budget (e.g., Trenberth et al., 2009; Wood, 2012). As such, it is of great importance to global climate studies to understand why there are two competing mechanisms reported in the literature regarding the pathways of LWP, which cause the LWP to either increase or decrease in response to an increase in aerosols.

Figure 2 shows the geographical distributions of LWP-susceptibility (i.e., the second term on the right side in Eq. 2) obtained from the MIROC simulation and satellite retrievals. The model produces positive values of $d \ln \mathrm{LWP} / d \ln N_c$ almost every-where over the globe in both non-precipitating and precipitating cases, which indicates that the LWP systematically increases with an increasing aerosol burden. Even when the model applies AOD or hygroscopic $N_a$ burden as an aerosol proxy instead of $N_c$, we obtain the similar results (i.e., globally enhanced LWP). This result is expected from the model parameterization of the cloud lifetime effect (Eq. 1), which monotonically delays the onset of precipitation in polluted conditions. This is also

a characteristic common to other GCMs, as reported in a recent study (Ghan et al., 2016). In contrast, the satellite-derived LWP-susceptibility has a coherent geographical pattern that includes both increasing and decreasing responses, which is quite different from the model results. The decreasing response occurs over the tropics and subtropics where more convective cloud is dominant. The increasing response is apparent mainly over the midlatitudes and regions where low clouds are dominant (Klein and Hartmann, 1993).

Nevertheless, it should be noted that Fig. 2c captures the horizontal distribution of LWP-susceptibility, whose pattern is very similar to observations. That is, the relationship becomes weaker towards the tropics, although the sign is still different. One of the possible mechanisms is the dominance of cloud dynamical processes with high natural variability over tropical/subtropical oceans rather than microphysical modifications by aerosols (Peters et al., 2011, 2014). The same processes observed from satellites could be at work in the model, and hence it might be related to the parameterization of subgrid-scale variability. However, this is not always true particularly in non-precipitating cases (Fig. 2a), so we must interpret the mechanisms carefully with further analysis in future.

Figures 2b and 2d show that the geographical patterns are qualitatively similar between the non-precipitating and precipitating conditions, whereas the contrast between the two is slightly different. More specifically, the value of $d \ln \text{LWP} / d \ln N_c$ is smaller in the precipitating condition than in the non-precipitating case, which implies a smaller effect of aerosols when precipitation occurs. However, it is noteworthy that the positive response of $d \ln \text{LWP} / d \ln N_c$ in the non-precipitating condition has a negative value in the precipitating conditions over East Asia, the eastern United States, and Europe, where the anthropogenic aerosol burden is severe. This suggests that aerosols act to prolong the cloud lifetime in non-precipitating conditions, while they enhance cloud evaporation or can be a precipitation driver in precipitating conditions which ultimately result in less cloud water. These two competing mechanisms are reasonably consistent with theories suggested by recent studies (Stevens and Feingold, 2009; Rosenfeld et al., 2014; Lebo and Feingold, 2014), which propose the existence of a buffering effect in the cloud system that results in smaller-magnitude aerosol–cloud interactions. These comparisons suggest that the model does not appropriately represent the buffering effect which compensates for the positive responses of the LWP to aerosol perturbations. Current GCMs which use similar parameterization framework (e.g., autoconversion) therefore inherently overestimate the aerosol indirect effect as reported by previous studies (Quaas et al., 2009; Wang et al., 2012; Gettelman et al., 2013).

Figure 3 summarizes the relationship among the "process-oriented metrics" corresponding to each term in Eq. (2). The global mean susceptibilities (averaged from 60°S to 60°N) for each term were calculated from the individual susceptibility in each grid box in which more than 10 warm cloud samples were obtained, which contributed to the reduction of statistical noise.

The cloud susceptibility of $\tau_c$ to $N_c$ in MIROC is approximately twice as large as in the A-Train results. The significant bias is decomposed into contributions due to the "albedo effect" $-d \ln r_e / d \ln N_c$ and the "lifetime effect" $d \ln \text{LWP} / d \ln N_c$. Although the former is underestimated in the model compared with A-Train estimations, its sign is positive and consistent with observations. The overestimation of $\tau_c$-susceptibility in the model is therefore attributed to a positive response of the LWP, which is in stark contrast to the slight negative responses in satellite observations. Ghan et al. (2016) reported a wide diversity in the relationship between the LWP and $N_c$ among nine AeroCom GCMs, with all models showing an enhanced response of

LWP to increased $N_c$. This further causes uncertainties in the estimation of radiative forcing (Ghan et al., 2016; Feingold et al., 2016).

As Figs. 2 and 3 indicate, the discrepancy in the LWP response between the model and observations can be a critical source of model uncertainty, which causes a bias in climate responses via the aerosol–cloud–precipitation–climate interaction.

## 3.3 Dependency of LWP responses on meteorology

Another key question regarding the LWP response for aerosol perturbations is how and to what extent it depends on macrophysics, such as cloud regimes and thermodynamic conditions in the real atmosphere. Recent studies have suggested that the source of uncertainty in the LWP response could be attributed to differences in meteorology, different cloud types and regimes, or more theoretical reasons, based on satellite observations (Sorooshian et al., 2013; Chen et al., 2014), large-eddy simulation (LES) (Lebo and Feingold, 2014; Seifert et al., 2015), and GCM intercomparison (Wang et al., 2012; Zhang et al., 2016).

To address this question, we examine the dependency of the LWP-susceptibility on both column maximum radar reflectivity ($Z_{max}$) and LTS as shown in Fig. 4. Given that the horizontal axis characterizes the rain regime (i.e., non-precipitating, drizzling, or precipitating) and the vertical axis represents the thermodynamical stability conditions (i.e., unstable, intermediate, or stable), the diagram illustrates how the LWP response to perturbed aerosols varies as a function of both rain characteristics and stability conditions; thus, providing a way to classify cloud susceptibility according to macroscopic, meteorological conditions.

Figure 4 clearly shows a systematic variation of the cloud susceptibility under the two conditions. Positive responses of the LWP to $N_c$ are dominant in the non-precipitating and stable environments, while negative responses can be seen in precipitating and unstable conditions. The top-left region in the diagram corresponds to a stratocumulus regime in the marine boundary layer. Because this type of cloud typically produces light precipitation (i.e., drizzle) (Wood, 2012) rather than heavy precipitation it depletes a large amount of cloud water, and the aerosols ingested into this type of cloud effectively act to enhance cloud water storage, resulting in a positive response of LWP to an increased aerosol loading. In contrast, the right-bottom region in the diagram corresponds to a more convective cumulus regime, which is present mainly over the tropics. This type of cloud is characterized by a relatively fast precipitation timescale (Sorooshian et al., 2013), and is favorable for cloud water evaporation due to the larger extent of entrainment mixing (Small et al., 2009), which results in negative responses of the LWP.

It is interesting that there is a positive correlation in the top-right region even though precipitation occurs. One possible interpretation of this is that the water vapor supply is dominant over the loss of cloud water by precipitation, and this type of cloud may correspond to the sustained frontal precipitation (precipitating nimbostratus) systems found mainly over midlatitude oceanic regions, where water vapor is abundant. In pristine/clean environments, which is referred to as "aerosol-limited" condition (Koren et al., 2014), aerosols ingested into clouds will tend to store the cloud water but also produce to more rain simultaneously due to abundant water mass. We note that it is just a speculation at this stage, and it might be related to background aerosol number and environmental conditions (cf. Sect. 4 for more discussion). It is also noteworthy that the bottom-left region displays negative susceptibilities even though precipitation does not occur. Non-precipitating clouds in a significantly unstable environment would correspond to inland/daytime cumulus. This tendency agrees with the results of a previous study (Small et al., 2009) that focused on the non-precipitating cumulus regime, and suggests a mechanism whereby

the LWP decreases with increased aerosol loading via evaporation–entrainment feedback. This results in a loss of cloud water without precipitation.

Although the model version of the LWP-susceptibility diagram is not shown, it will indicate positive value in the matrix overall, as is obvious from Fig. 2. The mechanisms proposed above must be confirmed by more detailed examinations using GCM and CRM with satellite simulators, or using high-resolution process modeling, such as LES, in future studies. However, the observation-based findings described above strongly suggest that rigorous studies focusing on macrophysical conditions, including regional characteristics of meteorological factors, in addition to microphysical conditions, are indispensable to better understand the response of the aerosol–cloud–precipitation interaction.

## 4   Summary and discussion

We explored the source of discrepancy in the aerosol–cloud–precipitation interaction for warm clouds between an aerosol–climate model and A-Train satellite retrieval. The instantaneous model output was analyzed using as many samples as possible to provide reliable statistics and fair comparisons with satellite observations.

We found critical biases in the model in the response of the LWP to aerosol perturbations. The model predicted a monotonic increase in the LWP over the globe, in contrast to the observations that clearly showed a regional variation of the LWP response that either increased or decreased with an increasing aerosol loading. This variability in cloud susceptibility observed by the A-Train was closely related to differences in meteorological factors, such as cloud regimes and thermodynamic conditions. For example, stratiform clouds under stable conditions had a tendency to increase the LWP given aerosol perturbations, while cumulus clouds over an unstable environment tended to decrease the LWP as the aerosol loading increased. The bidirectional responses of LWP (both positive and negative) found in satellite observations in different aerosol concentrations might be related to the concept of "optimal aerosol concentration ($N_{op}$)" recently suggested (Dagan et al., 2015a, b). More specifically, in case of $N_a < N_{op}$, clouds tend to be deeper with larger liquid mass as referred to as cloud invigoration (e.g., Koren et al., 2014) for increased aerosol loading, whereas the case of $N_a > N_{op}$ would be favorable for cloud suppression due to enhanced entrainment and evaporation. This could lead the bidirectional LWP-susceptibilities, although we cannot mention the exact mechanisms at this stage because $N_{op}$ also depends on both cloud geometric scale and environmental conditions (Koren et al., 2014; Dagan et al., 2015a, b) as well as aerosol types might be involved in.

This can explain why previous studies have reported conflicting results for the LWP response, with either an increase or decrease with increasing aerosol loading (e.g., Sekiguchi et al., 2003; Ackerman et al., 2004; Matsui et al., 2006). Previous studies have focused on different study regions and/or targets, which has resulted in different cloud responses due to the different mechanisms of aerosol–cloud–precipitation interaction. This means that global-mean cloud susceptibility is not very meaningful in constraining the aerosol–cloud–precipitation interaction. Future studies should consider the regional dependence of the susceptibility metrics (Terai et al., 2015).

The monotonic increase in the LWP with increasing aerosol loading in the model is attributed to the autoconversion scheme, which assumes only suppression of rainwater generation to account for the traditional cloud lifetime effect without its compen-

sation, and does not take meteorological conditions into account. Mechanisms that can decrease the LWP in polluted conditions, such as the enhancement of evaporation due to entrainment mixing (Small et al., 2009; Seifert et al., 2015), are not incorporated in our model. This means that the model fails to represent the buffering effect for the aerosol–cloud–precipitation interaction (Stevens and Feingold, 2009). Moreover, the model overestimates $S_{conv}$ around low LWPs compared with A-Train satellite retrievals due to uncertainties in process rates parameterization (Wood, 2005). This is evidence that the autoconversion in the model is too fast, which results in the LWP having a high dependency on the $N_c$. This bias in the model is consistent with a previous study that reported a higher LWP-susceptibility in GCMs due to their diagnostic treatment of rainwater (Wang et al., 2012).

In future studies, the aerosol–cloud–precipitation framework must be expanded to represent the effect of environmental conditions in a flexible manner, in addition to the microphysics. Current microphysical frameworks without such macrophysical aspects bring highly sensitive aerosol–cloud interactions, although they vary to some extent depending on the autoconversion scheme (Gettelman, 2015; Suzuki et al., 2015; Michibata and Takemura, 2015). Ghan et al. (2016) also reported a wide diversity in the LWP response to $N_c$ among various GCMs, and concluded that their inconsistency could mainly be attributed to their different representations of the autoconversion process. However, it is also true that different choices of microphysical scheme alone do not significantly improve the model biases in both cloud physics and cloud radiative effects (Michibata and Takemura, 2015), and these two requirements sometimes contradict each other (Suzuki et al., 2013a). This is due to the arbitrary nature of tuning and assumptions (e.g., artificial threshold parameters, diagnostic treatment of rain), which is a bottleneck in GCMs (Hoose et al., 2009; Quaas et al., 2009; Ghan et al., 2013). Recently, a fundamental model improvement was achieved by introducing a prognostic precipitation framework (Gettelman et al., 2015; Sant et al., 2015), which represents important progress in process representations for more realistic cloud and precipitation microphysics. This improvement is expected to overcome some of the common problems in GCMs, such as the overestimation of the aerosol indirect effect and spurious light rain (Walters et al., 2014; Gettelman et al., 2015; Sant et al., 2015).

Furthermore, a representation of subgrid-scale fluctuations has also been critical problem in GCMs. Although the magnitude as well as sign of LWP-susceptibility differs between the model and observations, the horizontal pattern is similar in precipitating conditions. The parameterization of subgrid-scale variability may partly contribute to weaken the aerosol roles by capturing the large natural variability of clouds especially over tropical/subtropical oceans (Peters et al., 2011, 2014), which would lead to more realistic representation of cloud dynamical processes. For example, Guo et al. (2011, 2015) showed that both positive and negative LWP responses can be represented in even a GCM framework, by the PDF-based macrophysics parameterization, called "Cloud Layers Unified By Binormals (CLUBB; Larson and Golaz, 2005)". Lebsock et al. (2013) estimated a weighting factor of process rate equations to consider the subgrid effects based on A-Train retrievals unless accretion process is significantly underestimated. The interaction between microphysics and subgrid-scale dynamics (microphysics–dynamics interactions) in GCMs is therefore one of the indispensable processes for incorporating buffering effects and for improving model physics as a whole.

Although this study focused only on warm-phase clouds, our findings regarding different cloud responses to aerosol perturbations between GCMs and satellite observations will assist future model development for more accurate climate simulations.

Further studies should also contain an extension of the research target from liquid to mixed/iced clouds, and from a process-level to a cloud system to understand the whole cloud system response to aerosol perturbations, taking into account the buffered system morphology.

*Author contributions.* T.M. proposed research; T.M. and T.T. designed research; T.M., T.T., K.S., and Y.S. performed research; T.T., K.S., and Y.S. contributed new analytic tools; T.M. analyzed data; and T.M. and K.S. wrote the paper.

*Competing interests.* The authors declare no conflict of interest.

*Acknowledgements.* This study was supported by JSPS KAKENHI Grant in Aid for Research Fellows (JP15J05544) and Scientific Research (JP15K12190), and the Environment Research and Technology Development Fund (S-12-3) of the Ministry of the Environment, Japan. K. Suzuki was supported by NOAA's Climate Program Office's Modeling, Analysis, Predictions, and Projections program with grant number NA15OAR4310153. Simulations by MIROC-SPRINTARS were executed with the SX-9/ACE supercomputer system of the National Institute for Environmental Studies, Japan. The original CloudSat data products were provided by the CloudSat Data Processing Center at CIRA/Colorado State University (http://www.cloudsat.cira.colostate.edu). The authors would like to express their heartfelt gratitude to the CloudSat science team. The CloudSat and MIROC-SPRINTARS climatological data for single-layer warm cloud used in this study can be obtained from the author, T. Michibata, via a direct request (michibata@riam.kyushu-u.ac.jp). Finally, the authors thank two anonymous reviewers for providing constructive suggestions and comments, as well as Karsten Peters for his insightful discussion, which have significantly improved the manuscript.

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

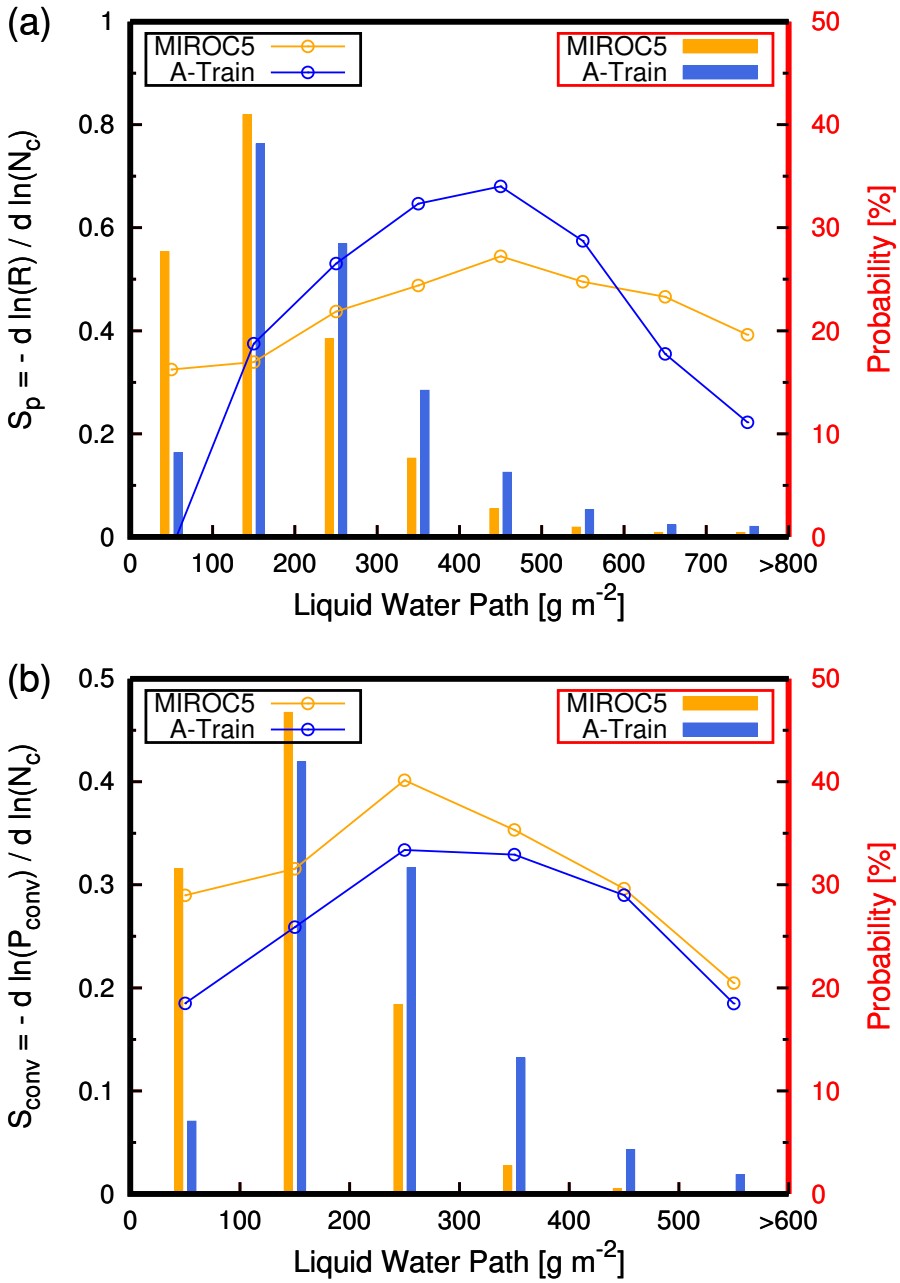

**Figure 1.** Susceptibilities of (a) precipitation $S_p$ and (b) microphysical conversion $S_{conv}$ as a function of the liquid water path (LWP) for the MIROC-SPRINTARS results and A-Train observations. The left axis shows the value of the susceptibility (refer to the line graph), and the right red axis shows the probability distribution function for each LWP bin (refer to the bar chart).

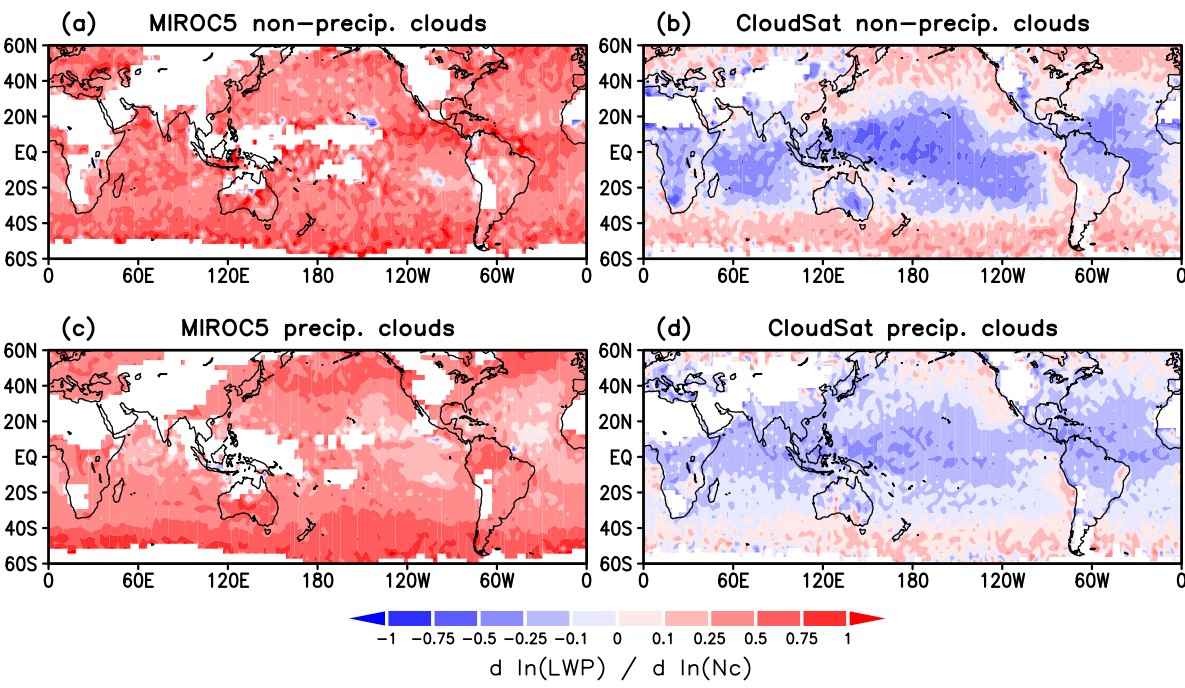

**Figure 2.** Global distribution of $d \ln \mathrm{LWP} / d \ln N_c$ from (a, c) MIROC-SPRINTARS and (b, d) A-Train satellite estimations for non-precipitating and precipitating clouds, respectively. The threshold of the large-scale precipitation rate of $0.14 \, \mathrm{mm \, day}^{-1}$ is used to distinguish between non-precipitating or precipitating events (see text for details).

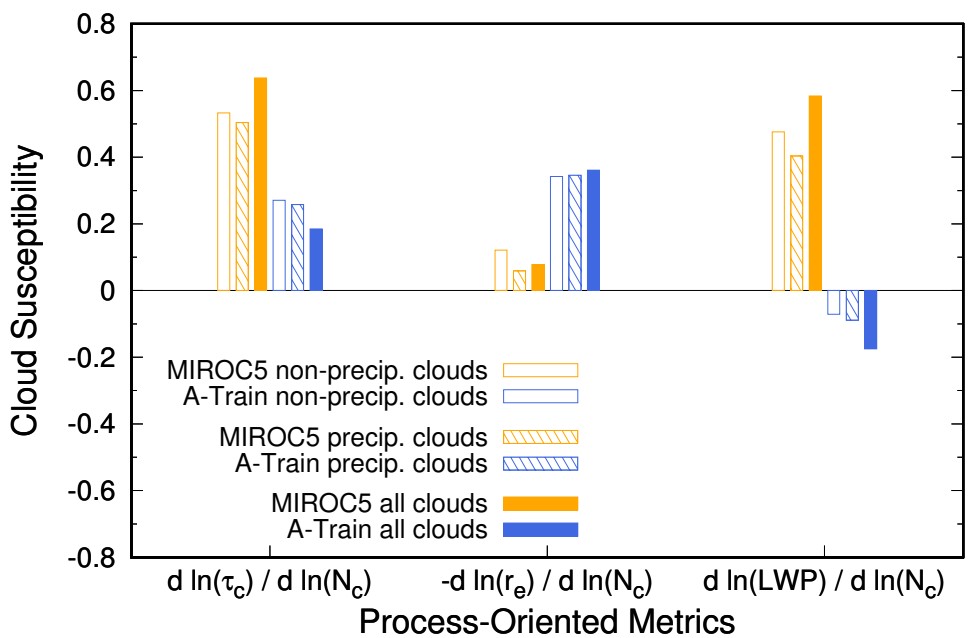

**Figure 3.** Global mean susceptibility (60°S–60°N) of $\tau_c$, $r_e$, and the LWP to $N_c$. The MIROC result is shown in orange and the A-Train observation is in blue.

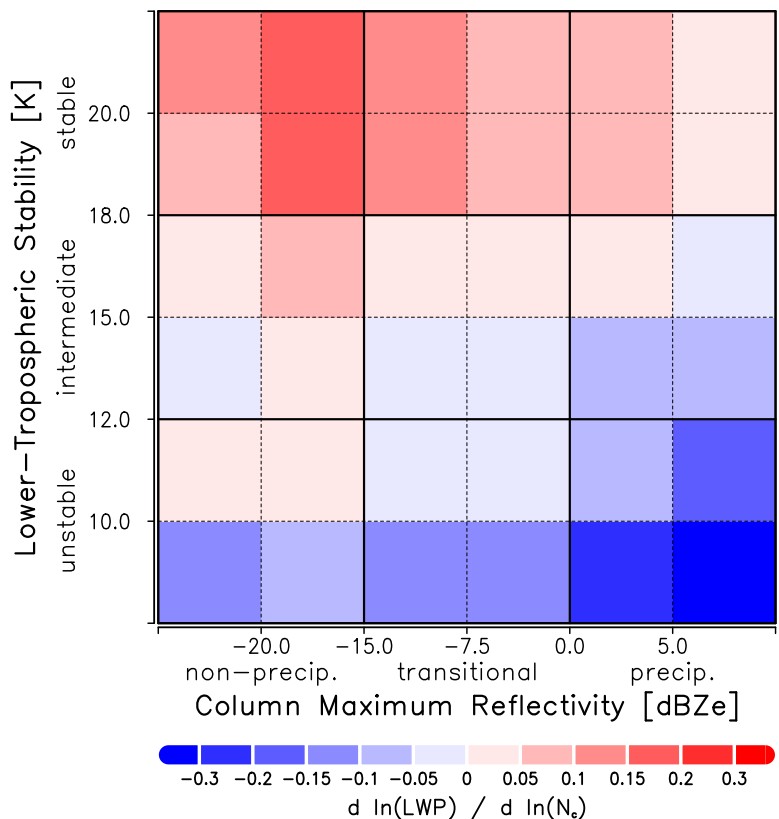

**Figure 4.** Susceptibility matrix of the LWP response to $N_c$ as a function of column maximum radar reflectivity ($Z_{max}$) and lower-tropospheric stability (LTS) based on A-Train satellite data.