# Peer review of "The source of discrepancies in aerosol–cloud–precipitation interactions between GCM and A-Train retrievals"

_Atmospheric Chemistry and Physics, 2016_

## Referee Comment (RC1) · Anonymous Referee #1 · 1 Oct 2016

This manuscript sheds light on the overestimate of aerosol effects on liquid water path simulated by GCMs. I particularly like the two-dimensional dependence on stability and reflectivity illustrated in Figure 4. The figures effectively illustrate the key points. The writing is generally lucid and concise. Only minor revision is needed before publication.

Minor comments.

Page 2, line 22. Replace "hydrometeor" with "geometric".

Page 5, lines 30-31. Couldn't the Sconv bias also be due to insufficient vertical resolution or biased cloud geometric thickness?

Page 6, line 4. "The response of cloud liquid water to aerosol perturbations determines the cloud lifetime". I think you mean cloud lifetime effect, although cloud fraction changes are also involved.

Page 8, line 32 – page 9, line 1. You might note that the overestimate in Sconv at low LWP might be partly due to insufficient dependence of autoconversion on LWP. See, e.g., Wood, JAS (2005).

Page 9, line 12. Replace "the assumption" with "assumptions".

---

## Referee Comment (RC2) · Anonymous Referee #2 · 11 Oct 2016

This paper investigates the strength of aerosol cloud interactions in both models and observations, seeking to examine the sources of the strong lifetime effect in the MIROC5 GCM. The authors show that the precipitation susceptibility for the model shows some similarities to satellite observations, but displays some different characteristics at low and high LWP, which they attribute to the autoconversion scheme in the model. They go on to show how the relationship between liquid water path (LWP) and cloud droplet number concentration (Nd) in the model and observations is very different, changing sign depending on the meteorological environment in the observations but not in the model. They suggest that this means that the precipitation scheme in the model is not capturing some important aspects of the precipitation process.

The paper is well written and the plots are appropriate. I think that this is a nice way of investigating the model and observational differences. There are a couple of points that I think need clarification, involving the possibility of correlated errors in the retrievals and the validity of the assumptions used in the satellite retrievals along with a few other small points. If these points are addressed, I feel this paper would be suitable for publication in Atmospheric Chemistry and Physics.

Specific comments

Sec 2.2: I am slightly concerned about the use of LWP and Nd from the same instrument and retrieval. Both of these are derived from the MODIS optical depth and effective radius retrievals, which themselves are retrieved together. This means that any errors in the retrieval of the effective radius or the optical depth will propagate through to the LWP and Nd, such that the errors in these derived properties are not independent. If the errors in the effective radius and optical are large enough, this can result in biases in the LWP-Nd relationship (the same thing also applies for the re-Nd relationship). Even random errors in the MODIS optical depth and effective radius retrievals would thus be able to generate a LWP-Nd or re-Nd sensitivity. These retrieval issues would not be replicated in the model output and could be part of the reason for the model-satellite discrepancy, especially in broken cloud regions.

P4 L14: It may also be important that the MODIS derived Nd and LWP depend on the adiabatic assumption, which is not valid in precipitating cases. Is it possible that the relationship in precipitating or broken cloud cases might be influenced by variations in the adiabaticity of the cloud? Again, this assumption would not affect the model results.

P5 L21: 'Pconv can be estimated' - it would make the paper a little more self contained if there was a brief description as to how. It looks like it is also connected to retrievals of the droplet number and cloud water content? Could this also be affected by correlated errors in the retrievals or are these CloudSat number and water content retrievals?

P5 L30: perhaps 'at a higher frequency ... compared to observations'

P5 L33: 'alternatively ... related to unrealistically light rain.' Just to check, the biases in condensation lead to lower LWP, which in turn leads to more light rain as the autoconversion rate is lower at low LWP?

P6 L22: Why is it more likely to find a change in the response of the relationship with precipitation in a high aerosol region? I would have thought that the LWP-Nd relationship is a property of the clouds rather than of the aerosols, which would make it relatively independent of the aerosol level as long as the LWP-Nd relationship is linear.

P7 L31: How difficult would it be to show the causes of the positive relationship at high stability in precipitating environments? It would help to demonstrate the dominant role of precipitation. At the moment, stability has almost as large an effect as precipitation but this does not fit so neatly into the explanation given (that precipitation is the driving factor in determining the strength of the LWP-Nd relationship).

Fig. 4: I understand that the model version of this figure will be positive almost everywhere, but is there still a pattern in the strength of the relationship that depends on stability or precipitation?

---

## Author Comment (AC1) · 18 Nov 2016

**Response to Reviewer #1 of acp-2016-831**

Dear Reviewer,

Thank you very much for taking your time to review our paper.
I am returning herewith a manuscript revised according to reviewers' comments.
I hope that the manuscript is now acceptable for publication in *ACP*.

[RC]: Referee comment in *Italic*
[AC]: Author comment

**General Comment:**

[RC] *This manuscript sheds light on the overestimate of aerosol effects on liquid water path simulated by GCMs. I particularly like the two-dimensional dependence on stability and reflectivity illustrated in Figure 4. The figures effectively illustrate the key points. The writing is generally lucid and concise. Only minor revision is needed before publication.*
[AC] We would like to thank the referee #1 for his/her very positive comments. The reply and corrections on individual issues are below.

**Minor Comments:**

[RC1] *Page 2, line 22. Replace "hydrometeor" with "geometric".*
[AC1] We have modified.

[RC2] *Page 5, lines 30-31. Couldn't the $S_{conv}$ bias also be due to insufficient vertical resolution or biased cloud geometric thickness?*
[AC2] The model vertical resolution in this study is 20 layers (sigma-pressure coordinate system: 0.995, 0.980, 0.950, 0.900, 0.830, 0.745, 0.650, 0.549, 0.454, 0.369, 0.295, 0.230, 0.175, 0.124, 0.085, 0.060, 0.045, 0.035, 0.025, and 0.008), which is finer near the surface. The modeled cloud geometric thickness is well represented quantitatively compared with observations (Fig. R1), but there are some biases quantitatively. This mainly stems from insufficient vertical resolution in model than CloudSat whose output is fine (~240 m by oversampling). It further causes biases in vertically integrated conversion rate $P_{conv}$, and also $S_{conv}$.

We add this issue in the revised version as follows: "Besides this, $S_{conv}$ can also be biased from the error of cloud geometric thickness due to insufficient vertical resolution in GCMs. In addition to the microphysical aspects mentioned above, biases in macrophysical structure are also related to model performances, which will be discussed later (cf. Sect. 3.3).".

[RC3] *Page 6, line 4. "The response of cloud liquid water to aerosol perturbations determines the cloud lifetime". I think you mean cloud lifetime effect, although cloud fraction changes are also involved.*
[AC3] This sentence has been modified as follows: "The response of cloud liquid water to aerosol perturbations determines the cloud lifetime via the modification of cloud fraction (Albrecht, 1989), and is thus related to global hydrological cycles as well as radiation budget (e.g., Trenberth et al., 2009; Wood, 2012).".

[RC4] *Page 8, line 32 – page 9, line 1. You might note that the overestimate in $S_{conv}$ at low LWP might be partly due to insufficient dependence of autoconversion on LWP. See, e.g., Wood, JAS (2005).*
[AC4] Yes, the dependence of autoconversion rate upon LWP and $N_c$ (i.e., ¥alpha and ¥beta in Eq. 1)

is also important issue. In this case, since Fig. 1b shows $S_{conv}$ behavior as a function of LWP, the bias in $S_{conv}$ relates to the dependence on $N_c$ rather than LWP. MIROC uses Berry (1968) autoconversion scheme, which is parameterized as ¥alpha = 3 and ¥beta = 1, and this microphysical dependency is consistent with Wood (2005) who suggested that ¥alpha ~ 2.8–3.0 and ¥beta ~ 1.4–1.5. However, it is also true that the dependence of autoconversion rate on LWP and $N_c$ remains a controversial issue in several literatures (e.g., Suzuki et al., 2013, 2015; Gettelman, 2015).

We slightly have modified the sentence as follows: "Moreover, the model overestimates $S_{conv}$ around low LWPs compared with A-Train satellite retrievals due to uncertainties in process rates parameterization (Wood, 2005).".

**[RC5]** *Page 9, line 12. Replace "the assumption" with "assumptions".*
**[AC5]** We have modified.

[Figure]

**Figure R1.** The (left) simulated and (right) satellite observed cloud geometric/hydrometeor thickness.

Thank you very much for reviewing our paper.

Sincerely yours,

Takuro Michibata

**References**

Albrecht, B. A.: Aerosols, cloud microphysics, and fractional cloudiness, *Science*, 245, 1227–1230, 1989.

Berry, E. X.: Modification of the Warm Rain Process, *Proc. 1st Conf. on Weather Modif.*, April 28–May 1, Am. Meteorol. Soc., Albany, New York, pp. 81–85, 1968.

Gettelman, A.: Putting the clouds back in aerosol–cloud interactions, *Atmos. Chem. Phys.*, 15, 12397–12411, doi:10.5194/acp-15-12397-2015, 2015.

Suzuki, K., Stephens, G. L., and Lebsock, M. D.: Aerosol effect on the warm rain formation process: Satellite observations and modeling, *J. Geophys. Res.*, 118, 170–184, doi:10.1029/2012JD018722, 2013.

Suzuki, K., Stephens, G., Bodas-Salcedo, A., Wang, M., Golaz, J.-C., Yokohata, T., and Koshiro, T.:

Evaluation of the warm rain formation process in global models with satellite observations, *J. Atmos. Sci.*, 72, 3996–4014, doi:10.1175/JAS-D-14-0265.1, 2015.

Trenberth, K. E., J. T. Fasullo, and J. Kiehl: Earth's global energy budget, *Bull. Am. Meteorol. Soc.*, 90311–90323, doi:10.1175/2008BAMS2634.1, 2009

Wood, R.: Drizzle in stratiform boundary layer clouds. Part II: Microphysical aspects, *J. Atmos. Sci.*, 62, 3034–3050. 2005

Wood, R.: Stratocumulus clouds, *Mon. Weather Rev.*, 140, 2373–2423, doi:10.1175/MWR-D-11-00121.1, 2012.

---

## Author Comment (AC2) · 18 Nov 2016

**Response to Reviewer #2 of acp-2016-831**

Dear Reviewer,

Thank you very much for taking your time to review our paper.
I am returning herewith a manuscript revised according to reviewers' comments.
I hope that the manuscript is now acceptable for publication in *ACP*.

**[RC]**: Referee comment in *Italic*
**[AC]**: Author comment

**General Comment:**

**[RC]** *This paper investigates the strength of aerosol cloud interactions in both models and observations, seeking to examine the sources of the strong lifetime effect in the MIROC5 GCM. The authors show that the precipitation susceptibility for the model shows some similarities to satellite observations, but displays some different characteristics at low and high LWP, which they attribute to the autoconversion scheme in the model. They go on to show how the relationship between liquid water path (LWP) and cloud droplet number concentration ($N_d$) in the model and observations is very different, changing sign depending on the meteorological environment in the observations but not in the model. They suggest that this means that the precipitation scheme in the model is not capturing some important aspects of the precipitation process.*

*The paper is well written and the plots are appropriate. I think that this is a nice way of investigating the model and observational differences. There are a couple of points that I think need clarification, involving the possibility of correlated errors in the retrievals and the validity of the assumptions used in the satellite retrievals along with a few other small points. If these points are addressed, I feel this paper would be suitable for publication in Atmospheric Chemistry and Physics.*

**[AC]** We would like to thank the referee #2 for his/her careful reading our manuscript and for giving positive suggestions.
We tried to revise our manuscript so as to answer to the comments.
Our reply and corrections on individual issues are below.

**Specific comments:**

**[RC1]** *Sec 2.2: I am slightly concerned about the use of LWP and $N_d$ from the same instrument and retrieval. Both of these are derived from the MODIS optical depth and effective radius retrievals, which themselves are retrieved together. This means that any errors in the retrieval of the effective radius or the optical depth will propagate through to the LWP and $N_d$, such that the errors in these derived properties are not independent. If the errors in the effective radius and optical are large enough, this can result in biases in the LWP-$N_d$ relationship (the same thing also applies for the $r_e$-$N_d$ relationship). Even random errors in the MODIS optical depth and effective radius retrievals would thus be able to generate a LWP-$N_d$ or $r_e$-$N_d$ sensitivity. These retrieval issues would not be replicated in the model output and could be part of the reason for the model-satellite discrepancy, especially in broken cloud regions.*

**[AC1]** The retrieval errors in satellite measurements of LWP and $N_c$ for different environmental conditions (e.g., cloud types and rain regimes) are one of the troublesome issues, which do not occur in models. The validity of an adiabatic assumption in their retrieval from satellites is also important issue. Although a use of satellite simulators is the best way for a fair comparison between satellite observations and model simulation, instead that we avoided the issue as much as possible by applying uncertainty thresholds for optical thickness ($< 5^{*}$) and effective radius ($< 1 \mu$m) as described in Sect.

2.2. As a result, 60.2 % of uncertain data were excluded by this procedure. These thresholds are very strict, and are helpful to reduce the $N_c$ uncertainty. Furthermore, we use logarithmic form (i.e., $d \ln X / d \ln N_c$, where $X \in$ tauc, $r_e$, LWP, $R$, and $P_{conv}$) rather than absolute value for constraining cloud and precipitation susceptibilities, which also contributes to reduce a sensitivity of them to the retrieval uncertainties (e.g., Feingold and Siebert, 2009; Sorooshian et al., 2009). We therefore think that the assumptions concerning about the satellite retrievals do not change our results and conclusion so much.

   We added notes for these issues in the revised manuscript as follows:
Page 4 Line 17: "We note that satellite data inherently include uncertainties stemming from retrieval assumptions, which are not replicated in the model output. Although it could be a part of reason for discrepancies between the model and observations, this would mostly be canceled when susceptibilities of cloud and precipitation to aerosol loading are evaluated by a logarithmic form.".
[*] the submitted discussion paper indicated uncertainty threshold of < 3 for optical thickness, but it was wrong. "< 5" is the right threshold value in this study, and we have corrected in the revised manuscript.

**[RC2]** *P4 L14: It may also be important that the MODIS derived $N_d$ and LWP depend on the adiabatic assumption, which is not valid in precipitating cases. Is it possible that the relationship in precipitating or broken cloud cases might be influenced by variations in the adiabaticity of the cloud? Again, this assumption would not affect the model results.*
**[AC2]** The authors agree with this concern. The note about uncertainties from satellite retrievals and assumptions has been added in the revised version as answered in **[AC1]**.

**[RC3]** *P5 L21: '$P_{conv}$ can be estimated' - it would make the paper a little more self contained if there was a brief description as to how. It looks like it is also connected to retrievals of the droplet number and cloud water content? Could this also be affected by correlated errors in the retrievals or are these CloudSat number and water content retrievals?*
**[AC3]** We added more detailed description with some references for derivation of the conversion rate $P_{conv}$ from satellites as follows:
Section 2.2: "To examine the cloud-to-rain conversion process, the conversion rate ($P_{conv}$) contributed from both autoconversion (collision–coalescence of cloud droplets) and accretion (collision of cloud droplets by raindrops) was derived from the approximation suggested by Stephens and Haynes (2007). This method is established by the continuous collection equation (Pruppacher and Klett, 1997) using observed drop size distributions. $P_{conv}$ was estimated from MODIS LWP and CloudSat mean cloud-layer radar reflectivity $\bar{Z}$ as
$$P_{conv} = c_1 \, \text{LWP} \, \bar{Z} \, H[Z - Z_c], \qquad (5)$$
where $c_1 = \kappa_2 / 2^6$ is a coefficient from collection kernel (Long, 1974) with $\kappa_2 = 1.9*10^{11}$ cm$^{-3}$ s$^{-1}$ and sixth moment factor with radar reflectivity. $H[Z - Z_c]$ is the Heaviside step function to exclude the cases that $\bar{Z}$ is less than critical radar threshold $Z_c$ of -15 dBZ for which conversion process is negligible (Matrosov et al., 2004). Although this formulation is based on marine stratocumulus cases from DYCOMS-II measurements (vanZanten et al., 2005), it is applicable for global analysis to study aerosol–cloud interactions (Stephens and Haynes, 2007; Sorooshian et al., 2013) in drizzling light rain cases ($\bar{Z} < 0 \, dBZ$). The parameterization and assumptions used in this method (Eq. 5) are also valid for comparison between observations and model simulation (Suzuki and Stephens, 2009). This brings valuable understanding for microphysical conversion processes and its timescales, which matches the scope of our study."
Page 6 Line 4–6: reconstructed in the revised manuscript.

**[RC4]** *P5 L30: perhaps 'at a higher frequency ... compared to observations'*
**[AC4]** We have modified, thanks.

**[RC5]** *P5 L33: 'alternatively ... related to unrealistically light rain.' Just to check, the biases in condensation lead to lower LWP, which in turn leads to more light rain as the autoconversion rate is lower at low LWP?*

**[AC5]** Yes. This sentence has slightly been modified as follows:
"Alternatively, it is also possible that the model has biases in the condensation processes, which lead to lower LWP and, thus, result in lower autoconversion rate.".

**[RC6]** *P6 L22: Why is it more likely to find a change in the response of the relationship with precipitation in a high aerosol region? I would have thought that the LWP-$N_d$ relationship is a property of the clouds rather than of the aerosols, which would make it relatively independent of the aerosol level as long as the LWP-$N_d$ relationship is linear.*
**[AC6]** This study applied to $N_c$ as an aerosol proxy instead of aerosol parameters (e.g., $N_a$, aerosol index, or AOD), because retrievals of the aerosol information are quite difficult and uncertain when cloud is also present in the retrieved profile simultaneously. Although a use of $N_c$ in observations also partly includes uncertainty due to the assumption of adiabaticity, this would mostly be disappeared when the LWP-$N_c$ relationship is evaluated by a logarithmic form, as answered in **[AC1]**. The validity of the use of $N_c$ as an aerosol proxy is supported by Chen et al. (2014) in observation-based study, and also in a modeling study we have confirmed that the similar results can be obtained even when the model applies AOD or hygroscopic $N_a$ burden as an aerosol proxy instead of $N_c$. This has been described in the discussion paper in Page 3 Line 1–3 and Page 6 Line 11–12.

The bidirectional responses of LWP (both positive and negative) found in satellite observations in different aerosol concentrations might be related to the concept of "optimal aerosol concentration ($N_{op}$)" recently suggested (Dagan et al., 2015a, 2015b). More specifically, in case of $N_a < N_{op}$, clouds tend to be deeper with larger liquid mass as referred to as cloud invigoration[*] (e.g., Koren et al., 2014) for increased aerosol loading, whereas the case of $N_a > N_{op}$ would be favorable for cloud suppression due to enhanced entrainment and evaporation. This could lead the bidirectional LWP-susceptibilities, although we cannot mention the exact mechanisms at this stage because $N_{op}$ also depends on both cloud geometric scale and environmental conditions (Koren et al., 2014; Dagan et al., 2015a, 2015b) as well as aerosol types might be involved in.

We have added the above discussion into Sect. 4 of the revised manuscript.
[*] We removed the sentence in Page 10 Line 4: "deepening cloud invigoration (Rosenfeld et al., 2014; Koren et al., 2014) and", because the mechanism of cloud invigoration works positive relationship between LWP and aerosol burdens in aerosol-limited conditions (Koren et al., 2014).

**[RC7]** *P7 L31: How difficult would it be to show the causes of the positive relationship at high stability in precipitating environments? It would help to demonstrate the dominant role of precipitation. At the moment, stability has almost as large an effect as precipitation but this does not fit so neatly into the explanation given (that precipitation is the driving factor in determining the strength of the LWP-$N_d$ relationship).*
**[AC7]** As referee pointed out, we recognize the importance of atmospheric stability in addition to the precipitation. The cloud dynamical processes that promote evaporation due to turbulent mixing are relatively small in high stability conditions whereas water vapor supply is abundant, which would lead on positive relation between LWP and aerosol loadings over midlatitude oceanic regions. In pristine/clean environments, which is referred to as "aerosol-limited" condition (Koren et al., 2014), aerosols ingested into clouds will tend to store the cloud water but also produce to more rain simultaneously due to abundant water mass. We note that it is just a speculation at this stage, and it might be related to background aerosol number and environmental conditions as discussed in **[AC6]**. We have partly added them in the revised manuscript (Page 8 Line 29–32).

The limitations of current remote sensing techniques, however, CloudSat or polar-orbit satellites measurements cannot capture the exact lifecycle with time-evolution explicitly because they scan instantaneous cloud–precipitation properties (i.e., snapshot), so it must be required high-resolution process modeling using LES or CRM for constraining the detailed mechanisms, as described in the last paragraph of Section 3.3. This will be addressed in future publication.

**[RC8]** *Fig. 4: I understand that the model version of this figure will be positive almost everywhere, but is there still a pattern in the strength of the relationship that depends on stability or precipitation?*
**[AC8]** Figure R1 shows the model version of the LWP-susceptibility matrix as a function of rain

regime and stability condition. We used stratiform precipitation rate that corresponds to radar reflectivity estimated from the *Z–R* relationship. As we noted, the model version shows positive LWP-susceptibility in the matrix overall, and the figure does not show the clear correlation of LWP-susceptibility on macrophysical regimes (rain intensity and atmospheric stability).

We further investigated the regional variations with some different environments (Fig. R2). The scatter plots of LWP-susceptibility in different regions from satellite shows positive relationship with LTS, whereas the model does not evident. This error means that the model misses microphysics–dynamics interactions. We added some suggestions for future model improvements, according to the *"short comment"* posted on the discussion forum of our paper. Please see the revised manuscript, Sect. 4.

[Figure]

**Figure R1.** Susceptibility matrix of the LWP response to $N_c$ as a function of stratiform precipitation rate and lower tropospheric stability (LTS) based on MIROC-SPRINTARS simulation.

[Figure]

**Figure R2.** LWP-susceptibility in different regions and rain regimes (black: non-precipitating clouds, blue: precipitating clouds) as a function of LTS from (a) MIROC-SPRINTARS simulation and (b) CloudSat-MODIS satellite observations.

Thank you very much for reviewing our paper.

Sincerely yours,

Takuro Michibata

*References*

Chen, Y.-C., Christensen, M. W., Stephens, G. L., and Seinfeld, J. H.: Satellite-based estimate of global aerosol–cloud radiative forcing by marine warm clouds, *Nature Geosci.*, 7, 643–646, doi:10.1038/ngeo2214, 2014.

Dagan, G., Koren, I., and Altaratz, O.: Aerosol effects on the timing of warm rain processes, *Geophys. Res. Lett.*, 42, 4590–4598, doi:10.1002/2015GL063839, 2015a.

Dagan, G., Koren, I., and Altaratz, O.: Competition between core and periphery-based processes in warm convective clouds – from invigoration to suppression, *Atmos. Chem. Phys.*, 15, 2749–2760, doi:10.5194/acp-15-2749-2015, 2015b.

Feingold, G. and Siebert, H.: Cloud–aerosol interactions from the micro to the cloud scale, MIT Press Cambridge, Mass, pp. 319–338, 2009.

Koren, I., Dagan, G., and Altaratz, O.: From aerosol-limited to invigoration of warm convective clouds, *Science*, 344, 1143–1146, doi:10.1126/science.1252595, 2014.

Long, A. B.: Solutions to the droplet collection equation for polynomial kernels, *J. Atmos. Sci.*, 31, 1040–1052, 1974.

Matrosov, S. Y., Uttal, T., and Hazen, D. A.: Evaluation of radar reflectivity–based estimates of water content in stratiform marine clouds, *J. Appl. Meteorol.*, 43, 405–419, doi:10.1175/1520-0450(2004)043<0405:EORREO>2.0.CO;2, 2004.

Pruppacher, H. R. and Klett, J. D.: Microphysics of clouds and precipitation, Kluwer Academic Publishers, 2nd edn., 954 pp., 1997.

Rosenfeld, D., Sherwood, S., Wood, R., and Donner, L.: Climate effects of aerosol-cloud interactions, *Science*, 343, 379–380, 2014.

Sorooshian, A., Feingold, G., Lebsock, M. D., Jiang, H., and Stephens, G. L.: On the precipitation susceptibility of clouds to aerosol perturbations, *Geophys. Res. Lett.*, 36, L13803, doi:10.1029/2009GL038993, 2009.

Sorooshian, A., Wang, Z., Feingold, G., and L'Ecuyer, T. S.: A satellite perspective on cloud water to rain water conversion rates and relationships with environmental conditions, *J. Geophys. Res.*, 118, 6643–6650, doi:10.1002/jgrd.50523, 2013.

Stephens, G. L. and Haynes, J. M.: Near global observations of the warm rain coalescence process, *Geophys. Res. Lett.*, 34, L20805, doi:10.1029/2007GL030259, 2007.

Suzuki, K. and Stephens, G. L.: Relationship between radar reflectivity and the time scale of warm rain formation in a global cloud-resolving model, *Atmos. Res.*, 92, 411–419, doi:10.1016/j.atmosres.2008.12.010, 2009.

vanZanten, M. C., Stevens, B., Vali, G., and Lenschow, D. H.: Observations of drizzle in nocturnal marine stratocumulus, *J. Atmos. Sci.*, 62, 88–106, doi:10.1175/JAS-3355.1, 2005.

---

## Author Comment (AC3) · 18 Nov 2016

**Response to Short Comment on acp-2016-831**

Dear Dr. Karsten Peters,

Thank you very much for posting the insightful and important comments on the discussion forum.
I am returning herewith a manuscript revised according to the comments.

[SC]: Short comment in *Italic*
[AC]: Author comment

**Short Comment:**

[SC] *In their submitted contribution to ACP, the authors investigate the reasons behind the discrepanies in cloud and precipitation response to changes in the number of cloud droplets $dN_C$ in observations (satellite) and an aerosol climate model (MIROC-SPRINTARS). By doing so, the cloud and precipitation response under conditions of changing aerosol concentrations is investigated (if a positive relationship between $dN_C$ and increasing aerosol concentrations is taken for granted). Overall, the authors find that the modelled sensitivity of cloud and precipitation responses to $dN_C$ are in disagreement with observations and that this disagreement most probably stems from the simplistic parameterization of autoconversion in the model. This has been known for quite some time now (see references in the submitted manuscript), therefore rendering the submission as yet another study showing the limitations of current generation aerosol-climate models to adequately reproduce observed aerosol-cloud interactions. The limitations of satellite observations for this purpose must also be kept in mind though. Unfortunately, the authors miss the opportunity to present at least some suggestions for future model improvements, which - given the wealth of data and diagnostics presented - would add significant punch to the submission.*

*In light of the above, I find the global distributions of d ln LWP / d ln $N_C$ shown in Figure 2c,d of the submitted manuscript very intriguing and investigating the shown relationships further would potentially add more substance to the science presented.*

*Although the magnitude and even the sign of the shown relationships in Fig. 2 differ significantly between observations and the model, the overall pattern is similar: the relationship becomes weaker towards the tropics - although still of wrong sign. The reason for this could be high natural variability and the dominance of cloud dynamical processes compared to microphysical ones (e.g. Peters et al. (2014)). The same processes could be at work in the model used in the present study.*

*From the model description presented in the manuscript, it appears that the prognostic cloud scheme used in MIROC-SPRINTARS accounts for subgrid-scale variability of clouds. If possible, it would be very interesting to investigate the response of cloud properties to $dN_C$ as a function of subgrid-scale variability as diagnosed in the cloud scheme. If there does exist a systematic relationship between cloud subgrid-scale variability and the cloud response to $dN_C$ in the model, such an analysis could provide important insights into the model physics and provide useful suggestions for improving the parameterisation of cloud microphysics.*

[AC] We would like to thank Dr. Karsten Peters for very insightful comments and suggestions for improving our manuscript.

Although Fig. 2a does not indicate a regional dependence of LWP responses clearly, Fig. 2c indeed shows the similar pattern of LWP-susceptibility to observations, that is, the relationship becomes weaker towards the tropics. One of the possible mechanisms is the dominance of cloud dynamical processes rather than microphysical modifications due to aerosol perturbations (Peters et al., 2011, 2014), hence it might be related to the handling of the subgrid-scale variability in the model.

Here we show the geographical distribution of PDF moments (variance and skewness) for total water content prognosed in MIROC (Fig. R1). This suggests that the spatial gradients of both variance

and skewness are larger in precipitating conditions (Figs. R1c and R1d) than in non-precipitating cases (Figs. R1a and R1b). The regions over tropics and subtropics where cumulus (inhomogeneous cloud) is dominant, show larger variance and strong positive skewness mainly due to the convective detrainment and/or dry air advection. These regions reasonably correspond to the area where LWP-susceptibility is relatively weaker. However, this is not always true particularly in non-precipitating cases, so we must interpret carefully the mechanisms with further analysis, which is beyond the scope of this paper.

For example, the spatial gradient of LWP and $N_c$ with different meteorology could also incur the spurious correlation of LWP-susceptibility due to their covariance (e.g., Grandey and Stier, 2010; Gryspeerdt et al., 2014). Alternatively, biases in cloud geometric thickness (i.e., dependence on vertical resolution) could cause fluctuations of the modeled LWP-susceptibility as well.

These obscures from the several possibilities mentioned above need to be clarified. This will require more detailed examinations of resolution dependence, regional characteristics using observations, and also some further sensitivity experiments in the PDF parameterization. Hence only limited materials are shown at this stage, and this issue will be investigated in future work.

We add some discussion about 1) a relationship between cloud subgrid-scale variability and modeled LWP-susceptibility and 2) future model improvements, according to the comments.

1) relationship between cloud subgrid-scale variability and modeled LWP-susceptibility
The following paragraph has been inserted in Section 3.2:
Page 7 Line 8–14: "Nevertheless, it should be noted that Fig. 2c captures the horizontal distribution of LWP-susceptibility, whose pattern is very similar to observations. That is, the relationship becomes weaker towards the tropics, although the sign is still different. One of the possible mechanisms is the dominance of cloud dynamical processes with high natural variability over tropical/subtropical oceans rather than microphysical modifications by aerosols (Peters et al., 2011, 2014). The same processes observed from satellites could be at work in the model, and hence it might be related to the parameterization of subgrid-scale variability. However, this is not always true particularly in non-precipitating cases (Fig. 2a), so we must interpret the mechanisms carefully with further analysis in future."

2) future model improvements
We believe that one of the most important future model developments is an introduction of prognostic precipitation framework in GCMs (e.g., Sant et al., 2015; Gettelman et al., 2015), as described in the discussion paper (Page 9, Line 4–17). In addition to this, the importance of the parameterization of subgrid-scale variability has been added in the revised manuscript as follows:
Page 10 Line 25–35: "Furthermore, a representation of subgrid-scale fluctuations has also been a critical problem in GCMs. Although the magnitude as well as sign of LWP-susceptibility differs between the model and observations, the horizontal pattern is similar in precipitating conditions. The parameterization of subgrid-scale variability may partly contribute to weaken the aerosol roles by capturing the large natural variability of clouds especially over tropical/subtropical oceans (Peters et al., 2011, 2014), which would lead to more realistic representation of cloud dynamical processes. For example, Guo et al. (2011, 2015) showed that both positive and negative LWP responses can be represented in even a GCM framework, by the PDF-based subgrid parameterization, called "Cloud Layers Unified By Binormals (CLUBB; Larson and Golaz, 2005)". Lebsock et al. (2013) estimated a weighting factor of process rate equations to consider the subgrid effects based on A-Train retrievals unless accretion process is significantly underestimated. The interaction between microphysics and subgrid-scale dynamics (microphysics–dynamics interactions) in GCMs is therefore one of the indispensable processes for incorporating buffering effects and for improving model physics as a whole."

[Figure]

**Figure R1.** Global distribution of PDF moments for total water content (at the cloud mid-level) prognosed in the MIROC-SPRINTARS subgrid scheme. (a, c) PDF variance and (b, d) PDF skewness are displayed for both non-precipitating and precipitating cases.

Thank you very much for insightful discussion and valuable comment for our paper.

Sincerely yours,

Takuro Michibata

*References*

Gettelman, A., Morrison, H., Santos, S., Bogenschutz, P., and Caldwell, P. M.: Advanced two-moment bulk microphysics for global models. Part II: Global model solutions and aerosol–cloud interactions, *J. Climate*, 28, 1288–1307, doi:10.1175/JCLI-D-14-00103.1, 2015.

Grandey, B. S. and Stier, P.: A critical look at spatial scale choices in satellite-based aerosol indirect effect studies, *Atmos. Chem. Phys.*, 10, 11459–11470, doi:10.5194/acp-10-11459-2010, 2010.

Gryspeerdt, E., Stier, P., and Partridge, D. G.: Satellite observations of cloud regime development: the role of aerosol processes, *Atmos. Chem. Phys.*, 14, 1141–1158, doi:10.5194/acp-14-1141-2014, 2014.

Guo, H., Golaz, J. C., and Donner, L. J.: Aerosol effects on stratocumulus water paths in a PDF-based parameterization, *Geophys. Res. Lett.*, 38, L17 808, doi:10.1029/2011GL048611, 2011.

Guo, H., Golaz, J.-C., Donner, L. J., Wyman, B., Zhao, M., and Ginoux, P.: CLUBB as a unified cloud parameterization: Opportunities and challenges, *Geophys. Res. Lett.*, 42, 4540–4547, doi:10.1002/2015GL063672, 2015.

Larson, V. E. and Golaz, J.-C.: Using probability density functions to derive consistent closure

relationships among higher-order moments, *Mon. Weather Rev.*, 133, 1023–1042, doi:10.1175/MWR2902.1, 2005.

Lebsock, M., Morrison, H., and Gettelman, A.: Microphysical implications of cloud-precipitation covariance derived from satellite remote sensing, *J. Geophys. Res.*, 118, 6521–6533, doi:10.1002/jgrd.50347, 2013.

Peters, K., Quaas, J., and Graßl, H.: A search for large-scale effects of ship emissions on clouds and radiation in satellite data, *J. Geophys. Res.*, 116, D24205, doi:10.1029/2011JD016531, 2011.

Peters, K., Quaas, J., Stier, P., and Graßl, H.: Processes limiting the emergence of detectable aerosol indirect effects on tropical warm clouds in global aerosol-climate model and satellite data, *Tellus B*, 66, 24054, doi:10.3402/tellusb.v66.24054, 2014.

Sant, V., Posselt, R., and Lohmann, U.: Prognostic precipitation with three liquid water classes in the ECHAM5–HAM GCM, *Atmos. Chem. Phys.*, 15, 8717–8738, doi:10.5194/acp-15-8717-2015, 2015.